# Managing Undernutrition in Pediatric Oncology: A Consensus Statement Developed Using the Delphi Method by the Polish Society for Clinical Nutrition of Children and the Polish Society of Pediatric Oncology and Hematology

**DOI:** 10.3390/nu16091327

**Published:** 2024-04-28

**Authors:** Agnieszka Budka-Chrzęszczyk, Agnieszka Szlagatys-Sidorkiewicz, Ewa Bień, Ninela Irga-Jaworska, Anna Borkowska, Małgorzata Anna Krawczyk, Katarzyna Popińska, Hanna Romanowska, Ewa Toporowska-Kowalska, Magdalena Świder, Jan Styczyński, Tomasz Szczepański, Janusz Książyk

**Affiliations:** 1Department of Pediatrics, Pediatric Gastroenterology, Allergology and Nutrition, Medical University of Gdansk, 80-210 Gdansk, Poland; agnieszka.szlagatys-sidorkiewicz@gumed.edu.pl (A.S.-S.);; 2Department of Pediatrics, Hematology and Oncology, Medical University of Gdansk, 80-210 Gdansk, Poland; 3Department of Pediatrics, Nutrition and Metabolic Diseases, The Children’s Memorial Health Institute, 04-730 Warsaw, Poland; 4Department of Pediatrics, Endocrinology, Diabetology, Metabolic Diseases and Cardiology of Developmental Age, Pomeranian Medical University, 71-252 Szczecin, Poland; 5Department of Pediatric Allergology, Gastroenterology and Nutrition, Medical University of Lodz, 91-738 Lodz, Poland; 6Department of Anesthesiology and Critical Care Medicine, Clinical Provincial Hospital No. 2 in Rzeszow, 35-301 Rzeszow, Poland; 7Department of Pediatric Hematology and Oncology, Collegium Medicum, Nicolaus Copernicus University Torun, 85-000 Bydgoszcz, Poland; 8Department of Pediatric Hematology and Oncology, Zabrze, Medical University of Silesia, 40-752 Katowice, Poland

**Keywords:** childhood cancer, nutrition, nutritional support

## Abstract

“Managing Undernutrition in Pediatric Oncology” is a collaborative consensus statement of the Polish Society for Clinical Nutrition of Children and the Polish Society of Pediatric Oncology and Hematology. The early identification and accurate management of malnutrition in children receiving anticancer treatment are crucial components to integrate into comprehensive medical care. Given the scarcity of high-quality literature on this topic, a consensus statement process was chosen over other approaches, such as guidelines, to provide comprehensive recommendations. Nevertheless, an extensive literature review using the PubMed database was conducted. The following terms, namely pediatric, childhood, cancer, pediatric oncology, malnutrition, undernutrition, refeeding syndrome, nutritional support, and nutrition, were used. The consensus was reached through the Delphi method. Comprehensive recommendations aim to identify malnutrition early in children with cancer and optimize nutritional interventions in this group. The statement underscores the importance of baseline and ongoing assessments of nutritional status and the identification of the risk factors for malnutrition development, and it presents tools that can be used to achieve these goals. This consensus statement establishes a standardized approach to nutritional support, aiming to optimize outcomes in pediatric cancer patients.

## 1. Introduction

The nutritional status of children with malignancies is of crucial significance in terms of short-term consequences, such as tolerance of chemotherapy, response to chemotherapy and radiotherapy, drug metabolism, rate of treatment-related complications, and mortality and quality of life, as well as the long-term consequences—survivors’ growth, motor, cognitive, and neurologic development, body composition, bone maturation, and mineral density [1,2,3,4,5,6,7]. With an understanding of this, nutritional support should be an integral component of complex anticancer management. However, to date, there have been major differences in the methods of diagnostics and nutritional therapy applied in particular childhood cancer centers in Poland. This consensus statement, developed by the Polish Society for Clinical Nutrition of Children and the Polish Society of Pediatric Oncology and Hematology, aims to provide unified recommendations on most relevant aspects of nutritional support in children with malignancies in high-income countries. It aims to detect malnutrition in pediatric cancer patients early and optimize nutritional interventions within this population. It emphasizes the significance of assessments of nutritional status and identification of the risk factors for malnutrition development, and it presents practical tools applicable in daily clinical practice.

## 2. Materials and Methods

The literature review of the PubMed database with the use of keywords, including pediatric, childhood, cancer, pediatric oncology, malnutrition, undernutrition, refeeding syndrome (RS), nutritional support, and nutrition, was conducted from January to February 2023. Only clinical trials, meta-analysis, randomized controlled trials, reviews, and systematic reviews written in English were included. During the literature review, particular consideration was given to works published in the previous 5 years (January 2018 to December 2022).

A group of lead researchers (A.B.-C., A.S.-S., E.B., N.I.-J., M.A.K., J.K.) initially proposed an e-Delphi questionnaire with a set of recommendations based on experts’ opinions. The e-questionnaire with a set of recommendations and the first draft were sent out to the group of seven experts (A.B., K.P., H.R., E.T.-K., M.Ś., J.S., T.S.), according to the Delphi method for Round 1. During the first round, experts were asked to anonymously express their opinions and generate ideas on individual recommendations (available response options in e-questionnaire: agree/disagree/have additional comments—open-ended question). Experts were encouraged to share additional opinions regarding the content of the recommendations, as well as the entirety of the presented text. It was agreed that consensus would be reached when a recommendation was accepted by each of the experts. After having received the responses from Round 1, the answers were analyzed, and a summary of comments was shared with co-authors. After Round 1, a consensus was achieved on 10 out of 13 recommendations (100% agreement). Recommendations 1, 3, and 9 were modified accordingly. The second e-questionnaire, containing non-consensus issues and the Round 1 results, were sent out to the experts for Round 2. During Round 2, a consensus was achieved on all proposed recommendations (100% agreement). Due to limited data, recommendations were based on authors’ experiences and opinions. The final draft was sent to the experts for approval. The Delphi Rounds took place from October to December 2023.

## 3. Results

The authors present 13 recommendations, which apply to the evaluation of nutritional status, identification of risk groups of developing malnutrition, optimal diagnosis and management of malnutrition, as well as the prevention, causes, and diagnosis of refeeding syndrome. The recommendations are detailed in the following discussion.

### 3.1. Malnutrition in Pediatric Oncology

The World Health Organization (WHO) divides malnutrition into the following three groups of clinical conditions: undernutrition—which can be further categorized as stunting (when a patient’s height is low for their age), wasting (when a patient’s weight is low for their height), and underweight (when a patient’s weight is low for their age). The second group is micronutrient-related malnutrition. The last group includes overweight and obesity [8].

In a recent position statement from The European Society for Paediatric Gastroenterology Hepatology and Nutrition (ESPGHAN) Special Interest Group on Clinical Malnutrition by Hulst et al. [9], a new definition of disease-associated undernutrition has been proposed. It has been defined as a condition resulting from imbalanced nutrition or abnormal utilization of nutrients which causes clinically meaningful adverse effects on tissue function and/or body size/composition, with a subsequent impact on health outcomes. According to the proposition of Hulst et al., when the parameters listed below are <−2 z-score, moderate–severe acute malnutrition should be suspected: weight-for-age (WFA) [in infants], weight-for-height (WFH)/weight-for-length (WFL), body mass index (BMI) [in children ≥ 2 years of age], and mid upper arm circumference (MUAC). Also, in the case of serial measurements, growth velocity should be assessed, and a decline in WFA/WFH/WFL/BMI z-score, of ≥1 standard deviation (SD), can be indicative for disease-associated undernutrition [9]. It was also emphasized that meeting the above-mentioned criteria is only one aspect of assessing the patient’s nutritional status [9]. In the Polish population, it is recommended to use growth charts from the “OLA and OLAF Study” [10] in children > 3 years of age and the WHO growth charts below this age [11]. Growth charts proposed in the “OLA and OLAF Study” contain z-scores only for BMI, while weight and height norms are presented in the form of a table with percentiles. Charts for MUAC interpretation are also available [11,12].

Undernutrition is more common in children with solid tumors than in patients with leukemias. Its prevalence is estimated as 0–10% for children with leukemia, 20–50% with neuroblastoma, and 0–30% with other malignancies [13,14]. The causes of malnutrition in children with malignancies can be divided into three main groups: those connected with the patient, those connected with the disease itself, including the histological type of cancer, its location and stage, and those related to the types of anticancer treatment [1,15]. The main risk factors that can be present at the moment of cancer diagnosis are listed in Table 1. Meeting at least one criterion from Table 1 classifies the patient into the high risk of developing malnutrition group. This classification should be carried out for every patient at the moment of cancer diagnosis.

The clinical conditions strongly linked to malnutrition include sarcopenia and sarcopenic obesity. Sarcopenia was redefined in 2018 by the European Working Group on Sarcopenia in Older People (EWGSOP2) as a progressive and generalized skeletal muscle disorder associated with increased likelihood of adverse outcomes, including falls, fractures, physical disability, and mortality [18]. Importantly, EWGSOP2 has emphasized the significance of low muscle strength over low muscle mass as a more practical determinant of sarcopenia, which facilitates early diagnosis and therapy of this disorder in clinical practice. In children, there are no established and unified criteria for diagnosing sarcopenia and sarcopenic obesity. However, a growing body of research underlines the important role of these clinical phenomena in pediatric oncology. The loss of muscle mass in children with malignancies may be caused by the patient’s poor clinical condition, fatigue, nausea, pain, presence of intravenous lines, hospital environment, and reduced social activity, leading to lower patient physical activity [19].

In pediatric oncology, undernutrition, overnutrition, sarcopenic obesity, and sarcopenia are significant clinical issues, which influence all aspects of antitumor therapy. For organizational reasons, these recommendations focus on diagnosing and managing undernutrition. However, it is essential to develop guidelines for preventing and treating overnutrition, sarcopenia, and sarcopenic obesity in pediatric oncology.

#### 3.1.1. Assessment of the Nutritional Status of a Child with Cancer

Before initiating oncological treatment, it is crucial to provide a comprehensive assessment of the actual nutritional status of the patient. The following combination of different methods, termed ABCD, is valuable:

A. Anthropometric measures;

B. Biochemical tests;

C. Clinical evaluation;

D. Dietary assessment.

A. Anthropometric measures include weight, height, BMI, and MUAC measurements. Nonetheless, it is crucial to remember that BMI can be disturbed by multiple factors, such as tumor mass or fluid imbalance. Additionally, BMI does not reflect the body adipose tissue distribution, sarcopenia, and sarcopenic obesity. Consequently, patients presenting with these clinical conditions may be misdiagnosed when assessed only with BMI; therefore, the other points of the nutritional status assessment need to be provided, including bioimpedance analysis, if possible.B. Biochemistry tests do not provide reliable markers of undernutrition. However, they may help describe the patient’s metabolic status. Due to some specific features of the treatment in pediatric oncology, the widely used methods of assessing the nutritional status are not suitable for children with cancer. The treatment and its side effects can easily affect the levels of glucose, ferritin, albumin, total serum protein, and lipid profile [20,21,22]. The incorrect results of these tests cannot be the only indication to start or stop nutritional support. They need to be interpreted with caution, and the full clinical context must be considered.

Therefore, we recommend regular monitoring of selected laboratory tests, including total serum protein, albumin, glucose, and electrolytes. Their concentration can be changed not only by the poor nutritional status, but also by the cancer itself and its intensive treatment [23,24]. Chemotherapy and antifungal treatment can cause significant electrolyte disturbances, due to the nephrotoxicity of the drugs used or gastrointestinal loses [23,24]. In malnourished patients, an increase in albumin and total protein concentrations may be the indicators of the effectiveness of nutritional intervention [25]. In clinical practice, regular monitoring of albumin levels and electrolytes not only results from the assessment of the patient’s nutritional status, but also from the need to assess the presence of potential treatment and cancer complications.

Also, vitamin D levels should be checked regularly to adjust its supplementation dosage according to the patient’s needs. It has been shown that vitamin D deficiency is common in the population of children undergoing treatment for cancer [26]. If possible, the assessment of the concentrations of thiamine, transferrin, retinol-binding protein, zinc, selenium, cobalamin, riboflavin, and vitamins A and E can be considered, but the authors are aware that these laboratory tests are not still the standard of care. In-depth insights into supplementation in pediatric oncology exceed the scope of this work. However, we recommend referring to the work by Podpeskar et al. for further information [27].

Biochemical tests play a significant role in monitoring nutritional therapy and identifying potential complications that may arise [28]. When starting nutritional therapy, especially in the form of parenteral nutrition (PN), it is important to monitor its effectiveness and, above all, the presence of any potential complications, also through biochemical testing. It is advised to regularly monitor electrolytes, blood gases, serum glucose, total protein, albumin, creatinine, lipid profile, bilirubin, and aminotransferases, especially when PN is introduced [29].

C. Clinical evaluation includes active searching for signs of malnutrition (as shown in Figure 1) and identification of the clinical conditions that can lead to it. It is crucial to interview the patient in detail about gastrointestinal symptoms, such as stomachache, diarrhea, constipation, loss of appetite, dysphagia, nausea, or vomiting. During the physical examination, the severity of mucositis should be noted and the local treatment provided, even if the symptoms are mild.D. Dietary assessment—a dietitian consultation aims to evaluate the patient’s daily dietary habits and the quality of their diet. It should consist of a complete dietary interview, including a patient’s current intake, food hygiene (especially in patients with immunosuppression), food allergies, intolerances, and preferences.

#### 3.1.2. Assessment of the Risk of Developing Malnutrition

The classification of a patient into a risk group of developing malnutrition should be carried out as follows:at cancer diagnosis: identify the risk factors listed in Table 1.during the oncological treatment: provide the nutrition screening tool for childhood cancer (SCAN) assessment (Table 2).

Various tools are available for screening for malnutrition in a general pediatric population, such as the Screening Tool for Risk of Nutritional Status and Growth (Strong Kids), Simple Pediatric Nutritional Risk Score to identify children at risk of malnutrition (PNRS), Paediatric Yorkhill Malnutrition Score (PYMS), Screening Tool for the Assessment of Malnutrition in Paediatrics (STAMP), or SCAN [30,31]. When assessing the risk of malnutrition development in a child undergoing anticancer treatment, using a practical, efficient, and reliable scale is crucial. Among the scales mentioned above, the SCAN scale is specifically designed for pediatric patients undergoing anticancer therapy. Moreover, the scale is simple and easy to use. With a simple question (“Does the patient have any symptoms related to the gastrointestinal tract?”), the SCAN enables the identification of multiple issues and symptoms caused by the disease or anticancer therapy. Recurrent nausea, persistent vomiting, diarrhea, constipation, gastrointestinal obstruction or ileus (postoperative, drug-induced, or radiotherapy-related), xerostomia, dysphagia, odynophagia, stomatitis, and acute or chronic stomachache (gastroenteritis, incl. typhlitis, and acute pancreatitis) can be easily identified as potential risk factors of undernutrition. Despite the existing body of research being insufficient to declare the superiority of either scale, we recommend using SCAN. As a part of the clinical assessment of nutritional status, we believe that it is a valuable and practical scale. Nonetheless, further research is necessary on this issue.

#### 3.1.3. Indications for Starting Nutritional Support in a Child with Cancer

It is crucial to recognize the optimal moment when nutritional support should be initiated. Assessment of the presence of indications for the initiation of nutritional intervention is essential both before and during treatment (see Figure 2).

There are specific groups of children with cancer in whom it is advisable to consider immediate nutritional support following diagnosis, to prevent the worsening of the nutritional status. This applies to patients identified as high risk for developing malnutrition during the treatment and the patients who present undernutrition before initiating anticancer treatment. Early nutritional support is also vital for all infants, regardless of the presence of other risk factors, since an age of <1 year is an independent risk factor of undernutrition. Special attention is warranted for breastfed patients, due to the numerous benefits of breast milk, including essential nutrients and immune support. Breastfeeding is associated with lower rates of infectious diarrhea, acute otitis media, obesity, and diabetes in a child [32]. However, in cases of childhood cancer, breastfeeding may be ineffective due to various factors, such as the child’s condition preventing effective sucking at the breast, inflammation of the oral mucosa, nausea, vomiting, secondary lactose intolerance, and maternal loss of milk due to high stress levels. Despite this, it remains valuable for emotional bonding and comfort during treatment.

There are histologic types of cancer in which natural clinical courses or standard treatment are associated with a high risk of malnutrition, such as acute myeloid leukemia, high-risk acute lymphoblastic leukemia, and several malignant solid tumors. The risk of malnutrition in childhood solid tumors depends also on their stage and localization. Importantly, all patients diagnosed with relapse of cancer are considered at high risk of undernutrition. Also, anticancer treatment may directly or indirectly lead to undernutrition, causing various gastrointestinal complications. Surgical procedures can result in short bowel syndrome or ileus, while radiotherapy and intense multidrug chemotherapy may lead to severe side effects such as malabsorption, subileus, mucositis, nausea, vomiting, diarrhea, anorexia, and dysphagia. Early nutritional support should also be considered for patients who receive high-dose chemotherapy in a hematopoietic stem cell transplantation procedure or those who suffer from graft versus host disease.

Figure 2 presents the list of indications for initiating nutritional intervention during anticancer treatment, based on the recommendations of the ESPGHAN [33]. These criteria are based on anthropometric measurements, assessment of the growing rate, weight gain, and also on the assessment of the current intake. However, these recommendations apply to the general pediatric population. Due to very intense and ongoing anticancer treatment in children and the high dynamics in changing the patient’s condition, including nutritional status, it is worth considering the initiation of nutritional treatment at earlier stages than those mentioned. The use of early nutritional treatment can lead to a reduction in the risk of nutritional deterioration.

It should be noted that every case of observed weight loss or lack of weight gain (over longer periods of time, also with growth assessment) is an indication for starting nutritional treatment.

**Figure 2 nutrients-16-01327-f002:**
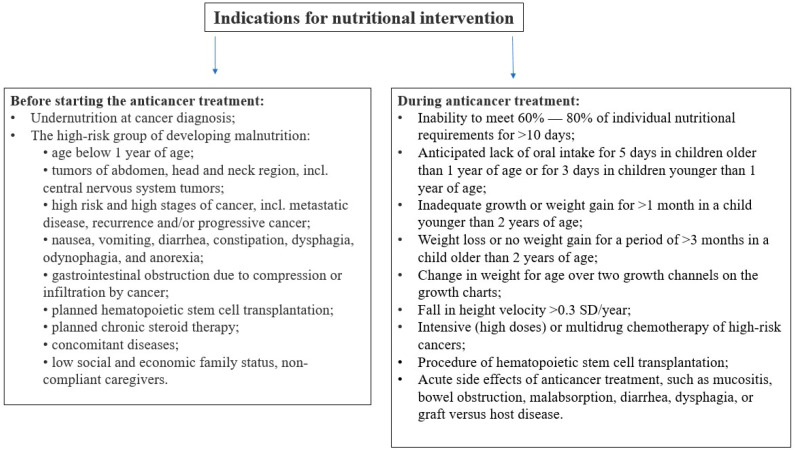
Indications for nutritional intervention (acc. to [16,17,33]; own elaboration).

#### 3.1.4. Frequency of the Assessments of the Nutritional Status

Regular assessments of nutritional status are mandatory in all cancer patients: at the time of cancer diagnosis, then at least every 2–4 weeks during intensive anticancer treatment, and at least every 4–8 weeks during maintenance therapy. Some patients require an individual approach, with the frequency of the assessments adjusted to their nutritional status at the time of cancer diagnosis, the initial risk of developing malnutrition, and current risk of malnutrition during anticancer treatment. For stable patients without signs of undernutrition, who have been tolerating oncologic treatment well, intervals for nutritional assessment may be extended.

### 3.2. Practical Aspects of Nutritional Support

The primary goal of nutritional intervention is to sustain and promote the child’s appropriate growth and development during anticancer treatment. Nutritional assessment and support are recommended in all children diagnosed with cancer to maintain proper nutritional status, restore existing nutritional abnormalities, or prevent future disorders that may occur during intense multimodal anticancer treatment. Thus, nutritional treatment should be an integral part of complex oncological treatment. Nutritional intervention should be implemented in case of insufficient dietary supply. It should be tailored to the particular clinical situation and should take into account the patient’s general condition, compliance, and current gastrointestinal tract function. It is crucial to discuss and explain the optimal available options for nutritional treatment with patients and caregivers, to ensure they understand the necessity and benefits of nutritional support.

The choice of nutritional intervention should be undertaken step-up-wise, with the intention that the more physiological a way of nutrition, the better (see Figure 3).

#### 3.2.1. Oral Route of Feeding and Enteral Nutrition (EN)

Due to a lack of data, there are no specific recommendations for providing a healthy oral diet for children undergoing anticancer treatment. It is considered that their everyday diet should not differ from that of a healthy child and should be adjusted to the patient’s age and nutritional needs. There are available recommendations concerning a healthy diet in children, although they primarily focus on guidelines for a healthy diet and obesity prevention [34,35]. In general, the diet should include whole grains, vegetables, fruits, lean meats, fish, dairy, plant-based protein, and fats sources. Consumption of sweets and sugary drinks should be minimized.

If a child experiences gastrointestinal symptoms, the diet should be individualized and modified. In the case of oral mucositis, dietary advice regarding modifications to the consistency, composition, and temperature of food (cold or at room temperature) is crucial for maintaining at least minimal oral food intake. In patients with vomiting and nausea, more frequent meals with smaller volumes may be proposed; some patients may benefit from a bland diet. Food at room temperature with minimal odor may be better tolerated. During infections, including gastrointestinal infections, due to the secondary malabsorption syndrome bland diet and lactose-free diet may be beneficial. In the case of post-steroid diabetes or impaired glucose tolerance, a diet with a low glycemic index is recommended.

Patients with neutropenia require special attention. A “neutropenic diet” that had been recommended previously has been proven not to reduce the risk of infection in this group of patients. Moreover, a neutropenic diet may even decrease the already limited oral intake of patients and pose unnecessary burden for parents and patients [36,37]. Nonetheless, in patients with neutropenia, food safety guidelines proposed by the Food and Drug Administration should be followed [37].

These special situations deserve attention, as they are everyday challenges faced in pediatric hemato-oncology wards. However, we considered this broad topic to exceed the planned scope of the presented recommendations. Children presenting gastrointestinal symptoms and metabolic disorders (hypertriglyceridemia, post-steroid diabetes) require regular dietary care.

The majority of patients who cannot meet their nutritional requirements through oral intake may benefit significantly from oral nutritional supplements and enteral nutrition.

Different types of intervention (oral/enteral/parenteral) may be combined to reach the nutritional goal. It is important to remember that EN does not exclude oral feeding, while parenteral nutrition (PN) does not exclude oral or EN. In pediatric oncology, multiple factors can disturb the functioning of the patient’s gastrointestinal tract and metabolism. They include the direct and indirect influence of the malignancy itself, as well as the side effects of various anticancer therapies, as shown in Figure 4. Therefore, it is crucial to carefully evaluate the current function of the gastrointestinal (GI) system to enable proper nutritional decisions. The decision algorithm, depending on the current function of the gastrointestinal tract in a cancer patient, is presented in Figure 5.

Oral route of feeding (personalized diet and/or oral nutritional supplements) should be preferred as the safest and most physiological way, in the maximal tolerated volume, to preserve gastrointestinal tract motility, proper digestion, and neurohormonal interplay.

For dietary counselling, periodical assessments of dietary supply should be carried out in all oncological patients, in terms of both quantity and quality. A personalized diet may be advised to meet the nutritional requirements. Dietary intake should be regularly monitored throughout the disease course and during subsequent therapies applied.

Oral nutritional supplements (ONS) should be introduced in patients who cannot meet nutritional requirements through a regular diet. ONS are designed to provide macro- and micronutrients orally and may be used as a sole source or as a supplementation of a regular diet [25,38]. ONS comprise liquid and powder formulas, which usually provide 1–2.5 kcal/mL. A range of products available on the market include juices, milk shakes, soups, or complete meals. Some of them have a neutral taste and may be used to prepare the child’s favorite meals.

Enteral nutrition is defined as the delivery of nutrients beyond the esophagus via the nasogastric or nasojejunal tubes (temporary GI tract access) or percutaneous tubes (permanent GI tract access) [33,39]. It may be provided to patients with at least a partially functioning gut, whose oral intake is insufficient to meet the current nutritional requirements.

A recent consensus statement by Fabozzi et al. proposes that the initiation of EN should be considered in all patients who are unable to meet 50% of their nutritional requirements with oral diet for more than 5 consecutive days [40].

There are many benefits of enteral nutrition for patients with cancer, including maintenance of intestinal mucosa integrity, neurohormonal interplay, provision of nutrients into enterocytes, prevention of bacterial translocation, and decreased risk of cholestasis. Compared to parenteral nutrition, the enteral route is cost-effective, safer and easy. If feasible and tolerated, even a minimal enteral diet should be provided during PN to protect the mucosal barrier and prevent enterogenic sepsis [41].

The absolute and relative digestive tract-related contraindications for EN are presented in Table 3. In case of relative contraindications, the enteral provision of the nutrients may be considered, weighing the risks and benefits of the route.

The types of accesses to the GI tract are presented in Figure 6. In most oncological patients, the gastric provision of nutrients via the nasogastric tube is sufficient and effective. Silicone or polyurethane tubes are recommended since they are flexible and delicate, easy to be accepted by the child and especially by the adolescent. Additionally, they need to be replaced after 4–6 weeks, while PVC tubes must be exchanged every 2–3 days. Permanent access to the GI tract should be considered if the anticipated duration of EN is longer than 4–6 weeks. However, every decision should be taken individually, considering the level of cooperation with the patient and caregivers, the acceptance of the method, the quality of life, and the risk and contraindications of gastrostomy insertion procedure under general anesthesia. Most commonly, gastrostomy can be inserted endoscopically. However, other methods (surgical, radiological) are also available. In the case of poor tolerance to gastric feeding or a high risk of food aspiration, gastroenterological consultation is recommended, as other GI tract accesses (duodenal/jejunal) might be required.

##### The Modes of Enteral Feeding via Nasogastric Tube

Enteral feeding via the nasogastric tube can be administered in the form of repeated boluses or intermittent or continuous infusion. Bolus feeding is the most convenient and physiological way of providing the diet via a nasogastric tube. However, in patients undergoing active intense anticancer treatment, it is recommended to start with continuous infusion (See Table 4). In the event of good tolerance, the transition from a continuous infusion to repeated boluses is advisable [40]. The equivalent formula administered through continuous infusion should be administered in divided doses every 3 h. Monitoring of the feeding tolerance is crucial. If EN is being resumed and the previously administered nutrition has been well-tolerated, the option of commencing directly with bolus feeding should be considered. The scheme of introducing enteral feeding via nasogastric tube in children has been outlined in Table 4.

In case the cancer patient is at risk of developing RS (was previously undernourished, experienced weight loss, underwent a significant reduction in nutrient intake, or manifested severe GI symptoms or treatment-related side effects, such as nausea, vomiting, mucositis, bowels obstruction, malabsorption, diarrhea, dysphagia, odynophagia, anorexia, or graft vs. host disease), please refer to the chapter “Refeeding syndrome” for the recommendations for children at risk of RS.

##### Choosing the Enteral Diet

Choosing the optimal enteral diet is implicated by many factors, such as the child’s age, nutritional requirement, volume tolerance, gastrointestinal function (motility, digestion, and absorption), and the need for a disease-specific diet (e.g., diets for diabetic, renal, or liver failure patients). Commercially available diets differ in composition, caloric density, and content of supplements (e.g., LCPUFA); see Figure 7 and Figure 8.

In terms of protein content, the diets can be classified as follows:polymeric—containing unmodified proteins (usually cow’s milk proteins), partially modified starch, and vegetable oils,oligomeric (semi-elemental)—with partially hydrolyzed proteins,monomeric (elemental)—containing proteins in the form of free amino acids, carbohydrates as disaccharides and glucose, and fat as long-chain triglycerides and medium-chain triglycerides.

The ESPGHAN recommends using enteral formulas instead of mixed regular diets in EN. The reasons include well-defined macro- and micronutrient compositions, adjusted to the patient’s current requirements, in a liquid form with low viscosity which is easy to provide via the nasogastric tube, sterility, comfort, and a lower complication rate [33]. However, the use of home-blended feeding is possible, provided that the safety rules of preparation and the correct composition of products are followed [42]. In the case of persistent signs of enteral nutrition intolerance, despite optimization of the type of formula and feeding regimen, a real food diet may be considered. Although research in this area is limited, there is growing evidence that it may be beneficial for this group of patients [43].

If there are no pre-existing limitations and contraindications, a complete polymeric normocaloric diet with fiber should be the first choice for the patient’s age. Further changes and modifications may be carried out according to the formula’s tolerance and side effects.

Most children can still eat while fed by the nasogastric tube. Usually, they accept the inserted tube within a few days—the smaller the child, the shorter the time is. Older children, as well as the parents, should be involved in deciding to insert the nasogastric tube, with a focus on properly understanding its benefits for better tolerance and outcome of oncological therapies. If long-term enteral nutrition is required, the procedure of home EN may be provided via the specialist home enteral nutrition centers.

#### 3.2.2. Parenteral Nutrition

The initiation of PN should be considered in patients who cannot meet nutritional requirements by oral or enteral routes. However, due to the possible complications of this method of nutrition and its non-physiological nature, the use of PN should be limited to a necessary minimum.

There are several options for intravenous access used for PN, including peripheral or central accesses. Among the central accesses, temporary and permanent accesses are available. Most cancer patients need the implantation of the permanent central venous access—a tunneled central catheter or vascular port. Any of the central venous accesses can be used also to provide PN; however, the use of vascular ports for the purpose to deliver PN is rare. It should be remembered that conducting PN through peripheral access is associated with certain limitations. Peripheral access is used primarily in patients who are expected to have a short duration of PN (up to 7–14 days), and temporary contradictions for implanting the central venous access exist in the patient. When conducting PN through peripheral access, the osmolality of the mixture should not exceed 900 mOsm/L [44], and the glucose concentration should be lower than 12% [29]. There is no need to eliminate lipid emulsions, since they are isotonic, so can be infused via both central and peripheral accesses [44].

The use of PN is associated with the risk of metabolic complications and those related to the venous access itself, such as the following: mechanical (e.g., hematoma, pneumothorax, damage to the venous or arterial vessels, nerve structures, air embolism, cardiac arrhythmia, or catheter dislodgement or damage), thrombotic, and infectious (e.g., catheter-related bloodstream infection) [44]. One of the metabolic complications of PN is abnormal glucose metabolism, including hypoglycemia and hyperglycemia. When introducing cyclic PN, it may be beneficial to measure blood glucose 30–60 min after the end of the infusion. In children with a tendency to hypoglycemia after the end of the infusion, a gradual reduction in the infusion rate may be needed [44]. The second metabolic complication could be hypertriglyceridemia. If triglycerides rise above 265 mg/dL in infants or above 400 mg/dL in older children, the reduction in the dosage of intravenous lipid emulsions is necessary [45]. Additionally, PN may be associated with the risk of the development of water–electrolyte imbalance, acid–base disorders, and refeeding syndrome. Patients who receive PN chronically may also develop metabolic bone disease or intestinal failure associated with liver disease [44].

##### Nutritional Requirements

Table 5 shows the basic energy, fluid, macronutrient, and macromineral requirements for children of each age group. However, it should be considered that these requirements may change depending on the patient’s clinical situation. For details on the management of PN, the choice of formulas, and mixture composition, see the ESPGHAN/ESPEN/ESPR/CSPEN guidelines on pediatric parenteral nutrition [46].

### 3.3. Refeeding Syndrome

Refeeding syndrome is an important and potentially life-threatening complication that can occur after any form of nutritional support—oral, enteral, or parenteral. It is characterized by an imbalance of electrolytes, particularly phosphorus, magnesium, potassium, and vital vitamins, especially thiamine, in blood. According to the American Society for Parenteral and Enteral Nutrition (ASPEN) Consensus [47], the diagnostic criteria for RS involve a reduction in serum phosphorus, potassium, or magnesium levels and multi-organ dysfunction resulting from a decrease in any of these or due to a thiamin deficiency (severe RS), which occur within five days of reinitiating or significantly increasing energy provision.

It is crucial to highlight that, in this specific population of children with malignancies, there is a risk of electrolyte imbalances resulting not solely from RS, but also from electrolyte losses via the gastrointestinal tract (diarrhea, vomiting, or fistulas) or via the urinary tract (tubulopathy or diuretic usage). Consequently, the clinical assessment of cancer patients demands a vigilant interview and cautious approach.

The incidence of RS, the risk factors of developing RS, and the management of RS have not been determined to date. However, the ASPEN consensus proposed the diagnostic criteria, clinical conditions likely to predispose patients to RS, and management strategies for RS, also in the pediatric population [47]. The criteria established by Silva et al. for recognizing individuals at risk for RS are not specifically tailored for pediatric patients undergoing anticancer treatment.

In this consensus, we propose to consider all children with cancer undergoing anticancer treatment as being at risk of developing RS, unless they were previously adequately nourished, have not experienced weight loss, have not undergone a significant reduction in nutrient intake, and have not shown any GI symptoms or treatment-related side effects (such as nausea, vomiting, mucositis, bowels obstruction, malabsorption, diarrhea, dysphagia, odynophagia, anorexia, or graft vs. host disease).

Manifestations of this syndrome can become apparent hours or days after the nutritional treatment is started [48]. The clinical presentation of RS is non-specific (see Table 6).

The proposition of guidelines for preventing and managing RS are available in the ASPEN Consensus [47]. In general, the following are advised:initiate nutritional support slowly;check electrolytes levels before the initiation of nutrition and supply deficiencies;regularly monitor electrolytes’ serum levels (in high-risk patients, every 12 h for the first three days of the nutritional treatment; in some patients, the frequency could be even higher, and need for monitoring could be longer);in high-risk patients, supply thiamine (2 mg/kg, max 100–200 mg/day) before initiating the nutrition and continue supplementation for 5–7 days (or longer);in high-risk patients, monitor vital signs every 4 h for the first 24 h;check the fluid balance and control patient’s weight daily.

The role of thiamine in oncology and the potential risks associated with its deficiency and supplementation remain unclear. Some studies have suggested a stimulating effect of thiamine on cancer cell proliferation, while others have shown its protective effects. Thiamine supplementation may even impact the effectiveness of cancer therapy. Conversely, there is evidence supporting thiamine’s protective role [49,50,51,52,53]. Due to the lack of sufficient data supporting the benefits or harms of thiamine supplementation, it was not possible to formulate recommendations regarding the supplementation of this vitamin in pediatric patients undergoing cancer treatment. The potential benefits and risks must be carefully assessed on an individual basis, taking into consideration the patient’s cancer type, treatment plan, and risk of developing RS.

In the case of starting the EN in patients at risk of developing RS, we advise initiating the treatment through continuous infusion, following the schedule outlined in Table 4. Should the continuous infusion be well-tolerated, transition to a scheme involving repeated boluses is recommended. In patients, in whom prior nutrition was well-tolerated and EN is being resumed, the possibility of initiating EN via bolus feeding may be taken into consideration.

We propose considering a gastroenterologist or nutritionist consultation before initiating nutritional support, in the case of a patient at risk of RS.

## 4. Discussion

The consensus statement, “Managing Undernutrition in Pediatric Oncology” developed by the Polish Society for Clinical Nutrition of Children and the Polish Society of Pediatric Oncology and Hematology, addresses the crucial aspect of proper assessment of nutritional status and the optimal management of undernutrition in children with cancer undergoing intense multimodal oncological treatment.

The importance of this topic has been associated with the evidence-based data that malnutrition can significantly negatively impact anticancer treatment tolerance and outcome and decrease the short- and long-term well-being of pediatric cancer patients. Therefore, nutritional treatment should become an integral component of medical care for pediatric patients with malignancies undergoing anticancer treatment. Proper nutrition is associated with better treatment outcomes, responses, and tolerance of chemotherapy, helps to alleviate side effects of treatment, improves the quality of life, and significantly impacts patients’ physical, motor, cognitive, and neurologic development [1,2,3,4,5,6,7].

To be able to maintain proper nutrition in a child with cancer, pediatric oncologists need to provide an accurate assessment of the patient’s nutritional status first. Therefore, this consensus statement proposes a customized nutritional status assessment scheme tailored specifically to this patients’ group. The statement aims to increase the frequency of diagnosing malnutrition and improve the nutritional status of patients by facilitating nutritional care. Our objective was to design an easy-to-use tool that will help to enhance nutritional treatment.

Several limitations warrant attention. Firstly, the topic of nutrition in pediatric oncology is still evolving. The body of research in this field is growing, but still needs to be improved. Due to various factors, including ethical considerations, large randomized controlled trials in cancer patients are lacking. The majority of trials are observational and retrospective. The literature search was conducted exclusively through one database (PubMed), which is also a limitation of the study. The consensus statement has been based on experts’ opinions and experiences, highlighting the necessity for further research to establish evidence-based practices in nutrition for pediatric oncology.

Secondly, our recommendations exclusively address the management of undernutrition in pediatric cancer patients, while other critical nutritional issues in pediatric oncology, including overnutrition and sarcopenic obesity, require the urgent development of guidelines. Additionally, specific anticancer treatment side effects (e.g., hypertriglyceridemia, glucose intolerance, post-steroid diabetes, neutropenia, and short bowel syndrome) require specific nutritional treatment.

To date, two studies have proposed nutritional recommendations for pediatric patients undergoing anticancer treatment: the International Society of Pediatric Oncology (SIOP) presented a framework for adapted nutritional therapy for children with cancer in low- and middle-income countries [54] and the Italian Association of Pediatric Hematology and Oncology, in collaboration with Survivorship Care and Nutritional Support Working Group of Alliance Against Cancer, developed a strategy for the management of nutritional needs in pediatric oncology [40].

Both studies emphasize the importance of nutritional therapy as a supportive treatment in oncology patients. They address, among other things, the tools used to assess nutritional status, the frequency of assessment, risk factors leading to nutritional disorders, and principles of enteral and parenteral nutrition. But the approach to the topic is different. Fabozzi et al. likely provides comprehensive guidelines and recommendations for managing nutritional needs, specifically in pediatric oncology patients. It is likely based on expert opinions and consensus among practitioners in the field. The report published by Ladas et al. emphasizes the challenges and considerations specific to low- and middle-income countries (LMICs), where resources for managing pediatric oncology patients, including nutritional support, may be limited. In summary, while both publications address the management of nutritional needs in pediatric oncology, they differ in their focus, scope, approach, and target audience, with Fabozzi et al. providing consensus-based recommendations and Ladas et al. offering a framework tailored for LMICs.

This consensus statement extends beyond the topics discussed in Ladas et al. and Fabozzi et al., addressing previously unmentioned issues and providing further elaboration on matters that our experts group deemed pivotal. The inclusion of experienced pediatric gastroenterologists, members of the Polish Society for Clinical Nutrition of Children, and pediatric oncologists from the Polish Society of Pediatric Oncology and Hematology signifies a diverse and specialized panel. The process of Delphi Rounds allows for the in-depth discussion and refinement of the proposed recommendations, ensuring that all viewpoints are considered and a consensus is reached among the panel members. This comprehensive approach addresses multiple facets of nutritional care, providing clinicians with practical guidance for managing the nutritional needs of pediatric oncology patients.

## 5. Recommendations

Recommendation 1.

To diagnose malnutrition, using the national growth charts and the WHO growth charts is recommended.

Recommendation 2.

Moderate–severe acute undernutrition should be suspected if WFA (in infants), WFH/WFL, BMI (in children ≥ 2 years of age]), or MUAC are equal to or lower than −2 SD, according to the norms set for the specific population. Growth velocity should be assessed and a decline in WFA/WFH/WFL/BMI z-score of ≥1 SD can be indicative of undernutrition.

Recommendation 3.

In all cancer patients, the child’s nutritional status, the risk of developing malnutrition, and the presence of indications for nutritional intervention should be assessed. This assessment is mandatory at the time of cancer diagnosis and then at least every 2–4 weeks during intensive anticancer treatment and at least every 4–8 weeks during maintenance therapy. However, in some patients, the frequency of nutritional assessments requires an individual approach.

Recommendation 4.

The assessment of nutritional status should include ABCD evaluation: anthropometric measures, biochemical (laboratory) tests, clinical evaluation of signs of inadequate intake and micronutrient deficiencies (see Figure 1), and dietitian consultation, as well as the evaluation of the presence of indications for nutritional intervention (see Figure 2), and, optionally, bioimpedance analysis.

Recommendation 5.

The classification of a patient into a high-risk group of developing malnutrition should be carried out as follows:at cancer diagnosis: identify the risk factors listed in Table 1.during the oncological treatment: provide the SCAN assessment.

Meeting at least one criterion from Table 1 or ≥3 points in the SCAN classifies a patient into a high-risk group for developing malnutrition.

Recommendation 6.

At the cancer diagnosis, nutritional treatment is recommended to each pediatric patient who has been found to be undernourished before therapy or who was assessed as being at a high risk of developing malnutrition during the oncological treatment.

Recommendation 7.

The nutritional intervention during anticancer treatment is recommended for any patient who meets at least one of the following criteria:is receiving intense multidrug chemotherapy for high-risk cancers (e.g., solid tumors in advanced stages, high grade brain tumors, high-risk acute lymphoblastic leukemia and acute myeloid leukemia, or cancer relapse);is being prepared for hematopoietic stem cell transplantation;shows side effects of anticancer treatment, such as mucositis, bowel obstruction, malabsorption, diarrhea, dysphagia, or graft versus host disease;is unable to meet 60–80% of individual nutritional requirements for >10 days;anticipated lack of oral intake for 5 days in children older than 1 year of age or for 3 days in children younger than 1 year of age;observed weight loss or lack of weight gain (over longer periods of time, also with growth assessment)—for details, see Figure 2.

Recommendation 8.

The nutritional intervention should be individualized and adjusted to the patient’s needs, clinical condition, and gastrointestinal tract function. The nutritional and energy requirements are similar to healthy children in stable, well-nourished patients at cancer diagnosis.

Recommendation 9.

More than one nutritional method can be used at the same time to achieve recommended nutritional provision. A step-up approach in decision making is recommended (see Figure 3).

Recommendation 10.

If oral nutrition (individualized diet +/− oral nutritional supplements) is insufficient to meet energy and nutritional requirements, enteral nutrition is recommended. As the first choice, a polymeric, normocaloric diet with fiber should be delivered via nasogastric tube (or gastrostomy, if present).

Recommendation 11.

In cases of enteral nutrition intolerance or unsatisfactory clinical effects, adjustments to the type of diet, as well as the mode and rate of its delivery, should be made.

Recommendation 12.

In the case of gastrointestinal tract dysfunction, parenteral nutrition should be considered. However, the duration of parenteral nutrition should be limited to a necessary minimum and simultaneous minimal enteral feeding should be given, if not contraindicated.

Recommendation 13.

Patients undergoing anticancer treatment should be considered as being at risk of developing RS, unless they were previously adequately nourished, have not experienced weight loss, have not undergone a significant reduction in nutrient intake, have not shown manifested GI symptoms or treatment-related side effects (such as nausea, vomiting, mucositis, bowels obstruction, malabsorption, diarrhea, dysphagia, odynophagia, anorexia, or graft vs. host disease).

## 6. Conclusions

Through the development of this consensus statement, our primary objective was to enhance the efficacy of nutritional support in pediatric oncology. We consider nutritional treatment a key element of comprehensive anticancer treatment.

## Figures and Tables

**Figure 1 nutrients-16-01327-f001:**
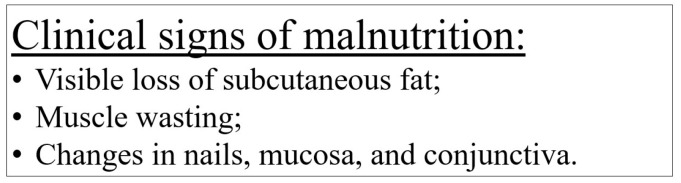
Clinical signs of malnutrition, own elaboration.

**Figure 3 nutrients-16-01327-f003:**
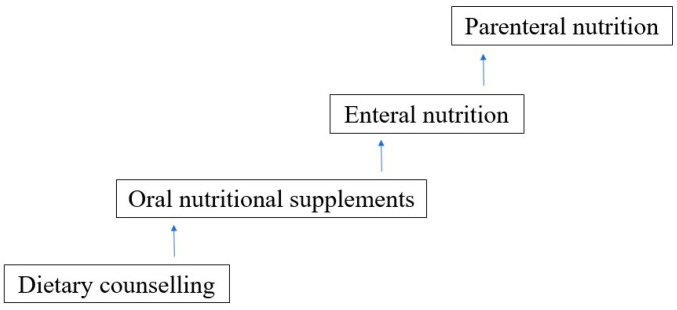
Nutritional intervention (step-up approach).

**Figure 4 nutrients-16-01327-f004:**
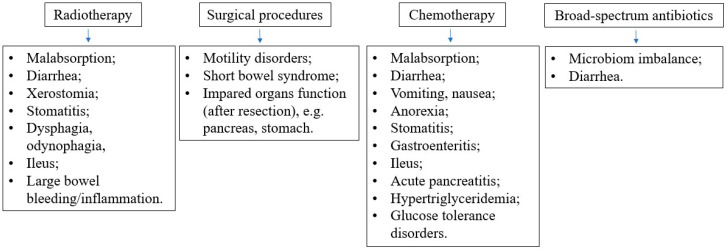
Side effects of the anticancer and supportive treatment associated with gastrointestinal tract and metabolic dysfunctions.

**Figure 5 nutrients-16-01327-f005:**
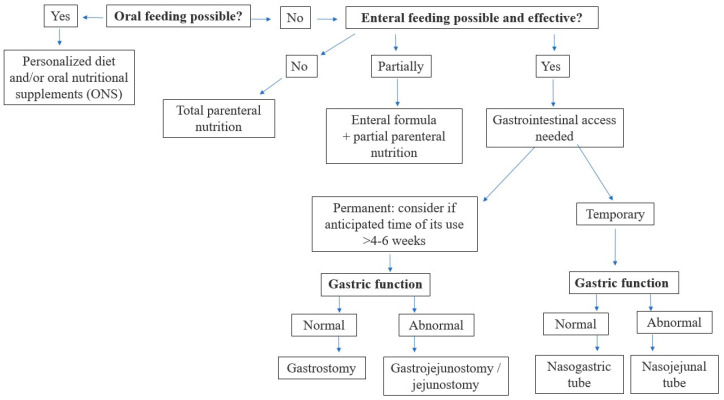
Decision algorithm for choosing the type of nutritional treatment concerning the current functioning of the patient’s gastrointestinal tract.

**Figure 6 nutrients-16-01327-f006:**
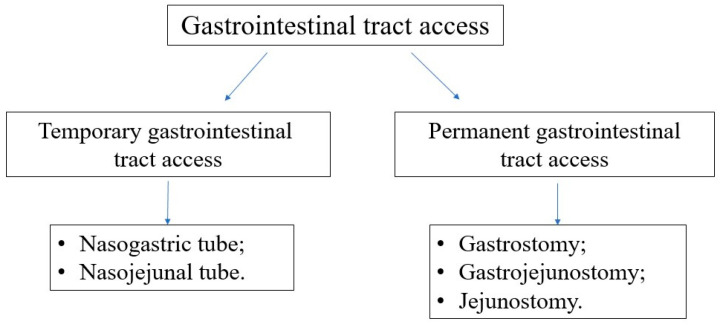
Types of gastrointestinal tract accesses.

**Figure 7 nutrients-16-01327-f007:**
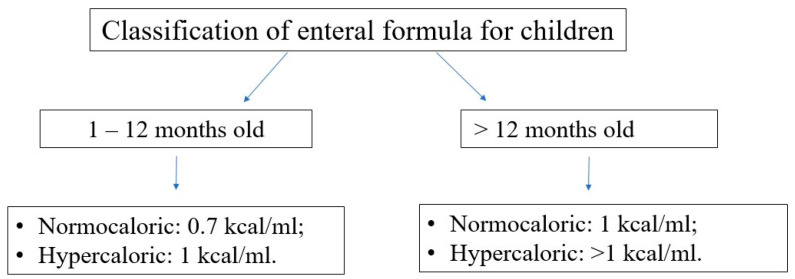
Classification of enteral formulas for children according to their age [33].

**Figure 8 nutrients-16-01327-f008:**
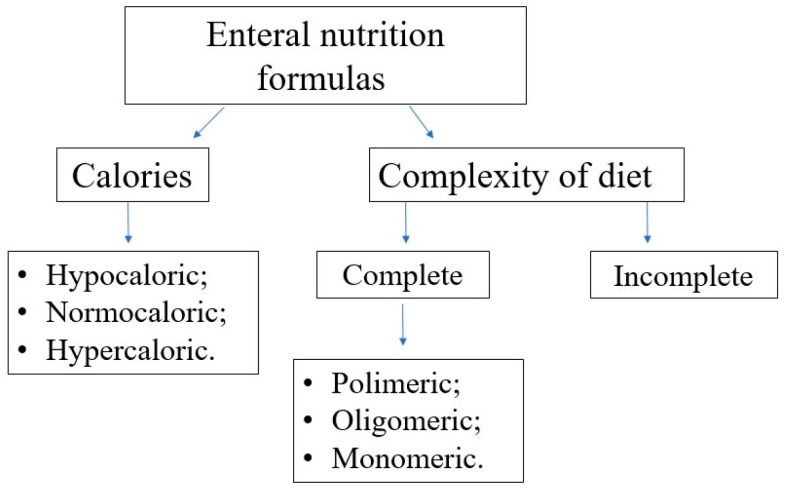
The enteral nutrition formulas regarding the calories and protein content.

**Table 1 nutrients-16-01327-t001:** Risk factors for developing malnutrition during the anticancer treatment; classification into the high-risk group (acc. to [16,17], own elaboration).

Risk Factors for Developing Malnutrition during the Anticancer Treatment
age below 1 year; malnutrition present at cancer diagnosis; tumors of abdomen, head and neck region, including central nervous system tumors; high risk and high stages of cancer, incl. metastatic disease, recurrent and/or progressive cancer; nausea, vomiting, diarrhea, constipation, dysphagia, odynophagia, and anorexia; gastrointestinal obstruction due to compression/infiltration by the tumor; planned hematopoietic stem cell transplantation; planned chronic steroid therapy; planned radiotherapy of abdomen and/or pelvis, head and neck regions; concomitant diseases; low social and economic family status, non-compliant caregiver.

**Table 2 nutrients-16-01327-t002:** Nutrition screening tool for childhood cancer (SCAN) [30].

Does the patient have a high-risk cancer?	1
Is the patient currently undergoing intensive treatment?	1
Does the patient have any symptoms related to the gastrointestinal tract?	2
Has the patient had poor intake over the past week?	2
Has the patient had any weight loss over the past month?	2
Does the patient show signs of undernutrition?	2
Score indication ≥3—at risk of malnutrition—refer to dietitian for further assessment	

**Table 3 nutrients-16-01327-t003:** Gastrointestinal tract-related contraindications for enteral nutrition [33].

Absolute Contraindications	Relative Contraindications
intestinal failure; complete intestinal obstruction; gut perforation; necrotic enterocolitis; inability to access the gut.	dysmotility; toxic megacolon; gastrointestinal bleeding; high output fistulas; peritonitis; persistent vomiting and diarrhea.

**Table 4 nutrients-16-01327-t004:** The scheme of the introduction of the enteral feeding via nasogastric tube in children.

**Continuous Infusion**	
Initial caloric intake per day	30% of recommended dose; 10–25% of recommended dose—in patients at risk of developing RS.
Time of continuous infusion	10–24 h—depends on a patient’s toleration.
Increase in caloric intake per 24 h *	30% of recommended dose; 10–15% of recommended dose—in patients at risk of developing RS.
**Repeated Boluses**	
Starting volume per feeding	10–30% of recommended dose; 10–25% of recommended dose—in patients at risk of developing RS.
Number of feedings per 24 h	<12 months: seven (every 3 h); >12 months: five–six.
Increase in caloric intake per feeding *	10–30% of recommended dose, if the residual volume before the next feed < 5 mL/kg or 200 mL in children > 40 kg; 10–15% of recommended dose, if the residual volume before the next feed is <5 mL/kg or 200 mL in children > 40 kg—in patients at risk of developing RS.
Full nutritional requirement	In 1–3 days; In 5–10 days—in patients at risk of developing RS.

* only if the previous volumes of enteral feedings have been well tolerated.

**Table 5 nutrients-16-01327-t005:** Basic nutritional requirements [29].

Basic Nutritional Requirements					
	28 days—12 m.o.	1–3 y.o.	3–7 y.o.	8–12 y.o.	13–18 y.o.
Fluids [mL/kg/day]	120–150	100–120	80–100	60–80	50–70
Amino acids [g/kg/day]	1.75–2	1.5–1.75	1.5–1.75	1.5–1.75	1–1.5
Glucose [g/kg/day]	10–17.5	10–12	10–12	7–10	5–7
Lipids [g/kg/day]	2–3	2–2.5	2–2.5	2–2.5	1.5–2
Na [mmol/kg/day]	1–2	1–1.5	1–1.5	1	1
K [mmol/kg/day]	1–2	1–1.5	1–1.5	1	1
Ca [mmol/kg/day]	0.6–0.8	0.2–0.4	0.2	0.2	0.2
P [mmol/kg/day]	0.6–0.7	0.3–0.4	0.2	0.1	0.1
Mg [mmol/kg/day]	0.2	0.15	0.1	0.1	0.1

**Table 6 nutrients-16-01327-t006:** Symptoms and laboratory markers of RS [47], own elaboration.

Cardiovascular System	Nervous System	Gastrointestinal System	Respiratory System	Other Symptoms	Abnormalities in Laboratory Tests
arrythmia; peripheral edema; pulmonary edema; cardiac failure.	weakness; paresthesia; polyneuropathy; qualitative and quantitative disorders of consciousness; paralysis; seizures; ataxia; ophthalmoplegia, nystagmus.	nausea; vomiting; constipation; anorexia; dysphagia.	respiratory failure; dyspnea.	rhabdomyolysis; sleep disorders.	hypokalemia; hypophosphatemia; hypomagnesemia; thiamine deficiency; lactic acidosis; thrombocytopenia; hemolysis.

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
