# Peer review of "Managing Undernutrition in Pediatric Oncology: A Consensus Statement Developed Using the Delphi Method by the Polish Society for Clinical Nutrition of Children and the Polish Society of Pediatric Oncology and Hematology"

_nutrients, 2024, doi:10.3390/nu16091327_

Round 1
Reviewer 1 Report (New Reviewer)
Comments and Suggestions for Authors
Thank you to the authors for this interesting article.
TITLE: you should state in the title that Delphi was the consensus methods used.
ABSTRACT:
Lines 30-32: this sentence should explain why a consensus statement process was chosen over other approaches (e.g. guidelines) referring to the paucity of high-quality literature regarding this topic.
Line 32: the fact that literature research was performed only through one database should be named as a study limitation.
Line 41 and line 585: to assert that “(…) recommendations are based on the 40 experts’ (…) experience” is a little bit redundant.
INTRODUCTION:
Lines 53-54: with the expression “(…) particular childhood cancer centers” do you refer to national or international centers? It would be better to define it.
Lines 54-57: does this consensus aim to have a national, regional, or global scope? You consider adding this information.
MATERIALS AND METHODS:
Lines 63-69: who performed the literature review? Did you use any instrument of clinical appraisal? Did you create any evidence report regarding the analyzed articles?
Lines 70-71: please explain who was responsible for the experts selection and report the criteria that were used to select the lead researchers and the panel of seven experts, providing the description of the role(s) and areas of expertise of all these subjects. You should also add the description of how experts were invited to participate in the consensus process.
Line 71: are the recommendations based only on experts’ opinion or also on the results of literature research? Did the experts receive any paper to read or any evidence synthesis to use as a support in defining recommendations?
Line 86: Here you assert that “Due to limited data, recommendations were based on authors’ experience and opinion.”, anyway in the text you cited at least two documents [55,41] that I guess can be considered as a meaningful source.
RESULTS:
Line 148: please correct the typing error “dernutrition”.
Author Response
Dear Sir/Madame,
I would like to express my sincere gratitude for dedicating your time to reviewing our manuscript and offering your insightful comments. Your feedback on our work is deeply appreciated, and we recognize the effort you invested in the review process. We have carefully considered each of your comments and acknowledge the importance of your input in refining our paper. We believe that your constructive criticism will significantly improve the quality of our work. Thank you once again for your thorough review. Below we send responses to the comments provided.
Best regards.

Reviewer 2 Report (New Reviewer)
Comments and Suggestions for Authors
Page 8, line 316 has a typing error. It should be 'dairy' not 'diary'.
Line 331, Grammar should be 'in patients with neutropenia'
More references should be used for this systemic review, 55 seem not sufficient for this manuscript.
Comments on the Quality of English LanguagePlease proofread the entire manuscript for grammar check and typing errors.
Author Response
Dear Reviewer,
We would like to express our sincere gratitude for providing feedback and for dedicating your time to reviewing our manuscript. Below, we are enclosing our responses.
Best regards.

Reviewer 3 Report (New Reviewer)
Comments and Suggestions for Authors
This exceptional article attempts to create much-needed nutritional guidelines for children with cancer. The PRISMA guidelines for systematic review are well followed, and the tables are informative. I do have some methodological remarks:
- Why did you search through only one database (instead of also Scopus, Cochrane,...).
- The Delphi method has the disadvantage that the criteria themselves are often not objective, and that these may be already leaning towards certain opinions. How did you cope with this?
Comments on the Quality of English LanguageSome minor typo's throughout the text.
Author Response
Dear Sir/Madame,
I would like to express my sincere gratitude for dedicating your time to reviewing our manuscript and offering your insightful comments. Your feedback on our work is deeply appreciated, and we recognize the effort you invested in the review process. We have carefully considered each of your comments and acknowledge the importance of your input in refining our paper. We believe that your constructive criticism will significantly improve the quality of our work. Thank you once again for your thorough review. Below we send responses to the comments provided.
Best regards

This manuscript is a resubmission of an earlier submission. The following is a list of the peer review reports and author responses from that submission.
Round 1
Reviewer 1 Report
Comments and Suggestions for Authors
the authors are to be applauded to embark on this difficult job, giving guidance on nutritional support in children with cancer. however, i do have serious doubts with the way things are described.
the authors describe that they had a 2-way proces: searching the literature for evidence, and secondly make recommendations by consensus. however, is not clear, thoughout the manuscript, what is based on evidence, and what is based on consensus
- most clinical guidelines nowadays include the opinion of parents/children, which is lacking here
- with respect on the evidence from literature: the search terms are presented, but a lot of information is lacking (from which date was searched, at which date the search was done, how many papers were found, which were in- and exclusion criteria, etc)
- with respect to the consensus recommendations; the authors describe "A group of lead researchers 63 initially proposed a set of recommendations and then sent it to co-authors according to 64 the Delphi Method. After having received the comments on each aspect from the co-au- 65 thors, the recommendations were modified accordingly." however, how was this done? how many Delphi rounds? were participants allowed to add recommendations, was there 100% consensus etc?
in all, the paper lacks transparency, which is crucial in modern guideline development.
more detailed remarks:
- in the discussion the authors describe that they focus on undernutrition. this should be clear from the title
- in line 96 the authors state "meeting at least one criterion....". however, furtheon they state that also scoring >2 on the SCAN should be considered. (line 188) please make consistent
- line 141: "undernutrition can be diangnosed..:" based on what? consensus, evidence?
- line 150 and further: on biochemisty: the authors advocate taking blood samples every 1-3 months. however, they do not describe why these measures are taken, what does it potentially cause, how are deficiencies in the general population, and what do you do with abnormal results. (even more taking into consideration their own remark on line 162)
- Line 178: i guess this should start with Add D.
- lien 256. all cancer patients need at least every 2 weeks .... i guess this does not apply for patients with very low risk treatment (eg ALL maintenance with 6mp/mtx, or maintenance therapy in eg rhabdomyosarcomas)
- the authors advice regular dietitian screening: for everyone? is that not a waste of time from the dieticians, who could better focus on those patients with serious needs. i presume the number of dieticians is also in Poland not unlimited.
- line 324: the authors refer here to a paper which is also a consensus guideline. this should be very explicit, since this is not "real evidence"
- line 367: explain abbreviation RS
- line 395: I miss the recommendation on the quatity of fibers in the enteral formulas
- line 499, please explain potential risks, other than RS
- line 596. recommendation 7: please add the high risk cancers.
- line 620, recommendation: sentence is not correct
Comments on the Quality of English Language
none
Author Response
Dear Sir/Madame,
I would like to express my sincere gratitude for dedicating your time to reviewing our manuscript and offering your insightful comments. Your feedback on our work is deeply appreciated, and we recognize the effort you invested in the review process. We have carefully considered each of your comments and acknowledge the importance of your input in refining our paper. We believe that your constructive criticism will significantly improve the quality of our work. Thank you once again for your thorough review.

Reviewer 2 Report
Comments and Suggestions for Authors
First of all we like to thank the authors for submitting this very important and interesting manuscript on nutritional support in children with malignancies.
Comments:
Overall, we recommend shortening the manuscript and emphasizing the structure more strongly for easier access to the information the reader is looking for.
Introduction: Please consider a short information on the current state of nutritional support in different childhood cancer centers – are there major differences? (line number 47 – 52) Please describe the objectives of the consensus statement here (line number 52 – 53) as in line number 535-536 and/or 635. In line number 535-536 the authors state they want to offer an “easy-in-use tool”. Therefore, it seems to be necessary to shorten the manuscript and to avoid duplications.
Methods: How did you realize exactly the Delphi method: were the experts' comments anonymous? (line number 63 – 68)
Results: Please consider offering a flow diagram for the literature search. It is unclear what role the literature research and experts’ opinions played and what their relationship was. (line number 70 – 74) The sentence “In the Polish population, it is recommended to use growth charts from the “OLA and OLAF Study” [10] in children >3 years of age and the WHO growth charts below this age [9].” is more or less the same in line 84 – 86 and line 136 – 138. The contents of figure 1 and the text (line number 168 – 169) are duplicated. Please avoid duplications (see above). In line 178 “Ad D.” is missing. The contents of the left half of figure 2 (page 6) are duplicated in table 2 (page 3). Again: please shorten as much as possible. In table 2 age under one year is mentioned as a risk factor. Please comment on the role of breastfeeding in this risk group. Furthermore, please explain a healthy diet for all the cancer patients in different situations e.g. infections and compare it to healthy children. In figure 4 (page 8) and in table 4 (page 10), the text in the description and the heading is the same, so the heading could be deleted. Please replace “disbalance” with “imbalance” in figure 4 (page 8). “Dietary counselling.” in line 309 isn’t a full sentence – consider a double dot.
You’re stating that “all children with cancer undergoing anti-cancer treatment as being at risk of developing RS” (line number 476 – 477). Therefore, you’re recommending, if necessary, enteral feeding via continuous infusion as shown in table 9 or via boluses as shown in table 10 (page 15). These are slightly different from table 5 and 6 (page 11). When would you recommend the latter as opposed to table 9 and 10? Please consider summarizing it in one table.
Discussion: Two comparable works are mentioned, please give a comparison with these – are there differences? (line number 550 – 557)
Author Response
Dear Sir/Madame,
I would like to express my sincere gratitude for taking the time to review our research paper and provide your insightful comments. Your feedback on our work is truly appreciated, and we value the effort you put into the review process. Each of your comments has been carefully considered, and we acknowledge the significance of your input in refining our paper. We believe that your constructive criticism will greatly enhance the quality of our work. Thank you once again for your thorough review.

Round 2
Reviewer 1 Report
Comments and Suggestions for Authors
my main problem with the manuscript is still the lack of transparency: what is based on consensus, what is based on evidence (and maybe even what is decided in a recommendation but is against (some) evidence.
if the authors wanted (as they say) to make a practical scheme what to do: that is OK, and worthwhile, but that should be distributed within the hospitals, and not presented as a manuscript
some additional remarks:
- page 4, ad B: "biochemistry tests do not provide reliable markers for undernutrition" however, i nthe next lined the authors recommend to use regular monitorin of seleceted laboratory rests. why?
and some lines further even: "biochemical tests play a significant role... etc, without references.
"figure 1: edema not related to fluid overload. is here evidence? because i think edema is much more frequently caused by capp. leak / inflammation, than by malnutrition
like in 1st review: why all assesments every 2-4 or 4-8 weeks. seems rather frequent for me. even when based on exp opinion, explanation is needed.
pate 18, recomm 5: meeting at least one criterion form table 2, or >= 3 in the SCAN: table 2 is about the scan.
page 19, recomm 7: high-risk alll and all. so? do the authors mean aml in the last case?
what is high risk all: if this is supposed to be a practical guideline, than it should be explicit (also on other places when they say high risk)
Author Response
Dear Reviewer,
We would like to express our sincere gratitude for providing additional feedback and for dedicating your time to reviewing our manuscript. Below, we are enclosing our responses.

Reviewer 2 Report
Comments and Suggestions for Authors
We would like to thank the authors for their revision of this important manuscript on the nutrition of paediatric oncology patients. Our suggestion to shorten the manuscript and thus really change it into an "easy-to-use tool" has been partially implemented. There is a spelling mistake in line 61 ("patietns"). The methodology is now more comprehensible. Breastfeeding is discussed from line 290 to 299. Please consider to shorten this new passage and give a little more information about the positive multifaceted value of breast milk and please add a reference on the latter, if possible. If possible, give a short comment (beside the given reference) on a fresh balanced diet; and how the diet should be adapted, e.g. for infections such as gastroenteritis, sepsis, etc.
